# Alkaline Electro-Sorption of Hydrogen Onto Nanoparticles of Pt, Pd, Pt_80_Pd_20_ and Cu(OH)_2_ Obtained by Pulsed Laser Ablation

**DOI:** 10.3390/nano13030561

**Published:** 2023-01-30

**Authors:** Antonino Scandurra, Valentina Iacono, Stefano Boscarino, Silvia Scalese, Maria Grazia Grimaldi, Francesco Ruffino

**Affiliations:** 1Department of Physics and Astronomy “Ettore Majorana”, University of Catania, via Santa Sofia 64, 95123 Catania, Italy; 2Institute for Microelectronics and Microsystems of National Research Council of Italy (CNR-IMM, Catania University Unit), via Santa Sofia 64, 95123 Catania, Italy; 3Research Unit of the University of Catania, National Interuniversity Consortium of Materials Science and Technology (INSTM-UdR of Catania), via S. Sofia 64, 95125 Catania, Italy; 4Institute for Microelectronics and Microsystems of National Research Council of Italy (CNR-IMM), Ottava Strada, 5 (Zona Industriale), 95121 Catania, Italy

**Keywords:** pulsed laser ablation, nanomaterials, copper hydroxide, hydrogen evolution reaction, alkaline media

## Abstract

Recently, hydrogen evolution reaction (HER) in alkaline media has received a renewed interest both in the fundamental research as well as in practical applications. Pulsed Laser Ablation in Liquid (PLAL) has been demonstrated as a very useful technique for the unconventional preparation of nanomaterials with amazing electro-catalyst properties toward HER, compared to those of nanomaterials prepared by conventional methods. In this paper, we compared the electro-sorption properties of hydrogen in alkaline media by Pt, Pd, Pt_80_Pd_20_, and Cu(OH)_2_ nanoparticles (NPs) prepared by PLAL. The NPs were placed onto graphene paper (GP). Noble metal particles have an almost spherical shape, whereas Cu(OH)_2_ presents a flower-bud-like shape, formed by very thin nanowalls. XPS analyses of Cu(OH)_2_ are compatible with a high co-ordination of Cu(II) centers by OH and H_2_O. A thin layer of perfluorosulfone ionomer placed onto the surface of nanoparticles (NPs) enhances their distribution on the surface of graphene paper (GP), thereby improving their electro-catalytic properties. The proposed mechanisms for hydrogen evolution reaction (HER) on noble metals and Cu(OH)_2_ are in line with the adsorption energies of H, OH, and H_2_O on the surfaces of Pt, Pd, and oxidized copper. A significant spillover mechanism was observed for the noble metals when supported by graphene paper. Cu(OH)_2_ prepared by PLAL shows a competitive efficiency toward HER that is attributed to its high hydrophilicity which, in turn, is due to the high co-ordination of Cu(II) centers in very thin Cu(OH)_2_ layers by OH^-^ and H_2_O. We propose the formation of an intermediate complex with water which can reduce the barrier energy of water adsorption and dissociation.

## 1. Introduction

The apparent rate of the hydrogen electro-sorption reaction depends on the physicochemical properties and according to the electronic structure of the specific electro-catalyst, as well as the pH of the solution. These factors must always be considered when interpreting the intermediate reaction steps, the adsorption of intermediates, the activation, and the barrier energies of the reaction [1].

In acidic solutions, the scientific community has accepted that the mechanism of hydrogen evolution reactions (HER) and their rate is mainly related to the hydrogen adsorption H_ads_ free energy exhibited on different materials [2,3,4,5,6,7]. Conversely, in alkaline solutions, the mechanism is still under debate in the scientific community [1]. 

Some authors maintain the bifunctional nature of HER in alkaline solutions and consider that the reaction is dependent on two main parameters, which are (1) the free energy of formal adsorption of hydroxyl, OH_ads_, which results from water adsorption and dissociation, and (2) the adsorption free energy of hydrogen, H_ads_ [8,9,10,11]. Platinum, palladium, Pt_80_Pd_20_, and their alloys, e.g., with silver, particularly in the size scale of nanometers, are high efficient electro-catalysts for the HER in acidic solution, because they have favorable energetics of H_ads_ [12,13,14,15,16,17]. However, the energetics of OH_ads_ are less favorable and represent the rate-limiting step in the HER in alkaline solutions. A successful strategy to increase the efficiency of HER in alkaline media consists of reducing the energy barrier of the water electro-sorption and dissociation on the surface of the electro-catalyst. Metals such as copper show poor HER activity, as reported by some authors [11,18]. Moreover, because copper is easily oxidized and hydroxidized, it presents a wide range of behavior toward HER in alkaline solutions. This fact is related to the different extent of surface oxidation presented by copper [19].

Dias Martins and coworkers proposed the formation of Cu(I)-OH_ads_ - - - OH_2_ activated complex as an essential intermediate for lowering the energy barrier for water dissociation onto a copper-oxidized surface. They correlate the surface oxidation extent, the oxophilicity, and the hydrophilicity with the catalytic activity for the HER, observing high activity for the HER in alkaline media [10].

Recently, we reported a study on the electro-sorption of hydrogen in alkaline solutions by Pt, Pd, and bimetallic Pt_80_Pd_20_ nanoparticles synthesized by Pulsed Laser Ablation in Liquid (PLAL), using pure water as a medium [20]. We found that these materials are still competitive in alkaline solution toward in relation to the HER, compared with other nanomaterials synthesized by conventional wet methods based on the chemical reduction of the precursors. We found that the presence of a thin layer of Nafion that surrounds the metal NPs further improves the efficiency of hydrogen electro-sorption by these NPs.

Furthermore, the electro-sorption efficiency depends upon the surface cleanliness of the electro-catalyst. In NPs produced by PLAL, the surface composition and purity can be controlled more accurately than that of an NPs’ surface produced by wet chemical reduction of a metal precursor. In particular, PLAL is suitable for the production of ligand-free surface NPs [21,22,23,24,25]. Several nanomaterials, produced by laser ablation, have been described in the literature as being suitable for the HER and storage [26]. In a typical experiment of PLAL, a nanosecond pulsed laser beam is focused onto the surface of a solid target placed in the specific liquid media. The absorbed radiation by the target produces an expanding plasma plume that contains the ablated material in the form of nanoparticle suspension [27]. Physicochemical, morphological, and size properties of NPs can be controlled by changing the laser parameters (wavelength, fluence, pulse duration) and/or the liquid media [28,29].

In this study, we compared the hydrogen electro-sorption activity in alkaline solutions using green nanoparticles of Pt, Pd, and Pt_80_Pd_20_, which are characterized by a favorable H_ads_ energetic—with high hydrophilicity, low cost, and non-critical material of Cu(OH)_2_— nanoparticles synthesized in a PLAL in-water environment. The effect of a thin layer of Nafion surrounding the NPs on the electro-sorption activity was also investigated.

## 2. Materials and Methods

### 2.1. Materials and NPs-GP Preparation

Graphene paper (GP) of a 240 μm thickness was used to fabricate the electrodes based on the PLAL NPs. GP presents several advantages in the electrode fabrication because it is lightweight and stable in alkaline solutions under a wide potential window. Moreover, it possesses a high electrical conductivity, similar to that of metals, and is characterized by an enhanced electron transfer at the electrode surface. GP, 5 wt.% solution of Nafion™ (sodium perfluoro-sulfonate ionomer), potassium hydroxide 99.99%, were purchased from Sigma Aldrich Merck (Milan, Italy). NPs suspensions of Pt, Pd, Pt_80_Pd_20_ (wt.%), and Cu(OH)_2_ were prepared by PLAL in water purified by a MilliQ™ system, characterized by electrical resistivity of 18.2 MΩ cm, and a total organic carbon content (TOC) of ≤ 5 part per billion (ppb). Pulsed Laser ablations (10 ns) were performed by Nd: Yttrium Aluminum Garnet YAG Laser operating at λ = 1064 nm, at fluence 5 J/cm^2^ and at repetition rate of 10 Hz, with 6W of power, using the Quanta-ray PRO-Series pulsed Nd:YAG laser (Spectra Physics, 1565 Barber Lane Milpitas, CA 95035 USA). Target of Pt, Pd, Pt_80_Pd_20_, and Cu with purity of 99.99% have been used. The duration of ablation was 8 min. More details of the experimental setup and the parameters used in the NPs preparation were reported in previous works [20,30]. Electrodes were fabricated using pieces of GP of 1 cm × 3 cm. The water-based NPs suspensions were then drop cast onto graphene paper (GP) using a hot plate at 100 °C, in an open-air environment, to cover an area of 1 cm^2^ on each side of the GP, resulting in the formation of NPs–GP nanoelectrodes. The electrodes were prepared by drop casting 29 μg cm^−2^ of platinum, or 5 μg cm^−2^ of palladium, or 5 μg cm^−2^ of Pt_80_Pd_20_ or 15 μg cm^−2^ of Cu(OH)_2_. Four sample batches were obtained, respectively. Water-0.25 wt.% of Nafion suspensions of NPs were employed to prepare a further set of electrodes, using the former process. A total of 200 μL of a 5 wt.% Nafion stock solution was added to 4 mL of each nanoparticle suspension. We estimated an average thickness of Nafion film surrounding the NPs of 0.7 μm [31].

### 2.2. Instrumental Characterization

A Gemini 152 Carl Zeiss Supra 25 and 35 field emission scanning electron microscopy (FE-SEM) (Jena, Germany), operating with the detector in the in-lens mode, were used to investigate the morphology of NPs–GP electrodes. Typical instrumental parameters were a working distance of 3 mm, a beam acceleration potential of 5 kV, and an aperture size of 30 μm. 

A PHI 5000 Versa Probe II system ULVAC-PHI, Inc. (2500 Hagisono, Chigasaki, Kanagawa, 253-8522, Japan) was used for the X-ray photoelectron spectroscopy (XPS) characterization. The spectra were excited by monochromatized Al Kα X-ray radiation. Moreover, a PHI 5600 multi technique ESCA-Auger spectrometer was used to analyze the Cu(OH)_2_ sample by Mg Kα X-ray radiation. Both Shirley and linear background subtractions were used to determine the XP peak intensities. C 1s main peak of graphene at 284.6 eV of binding energy was assumed for the binding energy scale calibration. Electrochemical measurements were performed in air at 22 °C by a potentiostat VersaSTAT 4 (Princeton Applied Research, 801 South Illinois Avenue, Oak Ridge, TN 37830 USA). In each measurement, 30 mL of fresh solution of KOH 1 M was used. The KOH solution was renewed in each measurement. The potential of the working electrodes was referenced to the Saturated Calomel Electrode (SCE). Platinum wire was used as counter. The electrochemical characterization of nanostructures was performed by Cyclic Voltammetry (CV) at a scan rate of 20 mVs^−1^, galvanostatic charge–discharge curved at a constant current of −100/+100 μA, and Electrochemical Impedance Spectroscopy (EIS) in galvanostatic mode at 250 μA cm^−2^ rms in the frequency range of 10^4^–1 Hz, respectively. A single charge or discharge cycle was performed in 300 s. Charge and discharge specific capacities of the NPs–GP systems have been calculated at a potential of −0.3 V and +0.25 V vs. SCE, respectively. Faradaic efficiency was calculated by the discharge-to-charge capacity percentage ratio. More details on the experimental procedures are reported in the Reference [20].

## 3. Results and Discussion

### 3.1. Morphology of NPs

Figure 1a–d report the morphology, studied by FE–SEM, of Pt, Pd Pt_80_Pd_20_, and Cu(OH)_2_ NPs, drop cast onto GP by the respective water-based PLAL suspensions. The noble-metal NPs have an almost spherical shape. The structures of these NPs were studied by XRD which revealed their crystalline nature [20,32]. The NPs have an average size of 8 to 16 nm [20,32]; however, some larger NPs may be observed in the FE–SEM pictures of Figure 1a–c, which are attributed to the tail of the size distribution of the NPs obtained by PLAL. According to our previous studies, the structure of Pt_80_Pd_20_ NPs is not of a core-shell type [20,32]. The morphology of the Cu(OH)_2_ NPs has the shape of flower buds with a size ranging from 200 to 400 nm. The NPs are formed by very thin nanowalls which are bound together. The structure of Cu(OH)_2_ NPs was studied by XRD and the results were reported in a previous paper [30]. XRD revealed the presence of Cu° and Cu_2_O species. These species are promptly oxidized and hydroxidized to Cu(OH)_2_. Figure 1e–h show the FE–SEM morphology of the NPs drop cast onto GP using water suspension containing 0.25% wt. of Nafion. The presence of Nafion reduces the agglomeration of the NPs. In fact, Nafion has numerous negatively charged sulphonic groups that cover the surface of the NPs and repel them. As a consequence, Nafion increases even more the extension of the NPs’ solution interface, which is beneficial for the electro-sorption reaction of hydrogen. 

### 3.2. Electronic Structure of NPs Surface

Figure 2 reports the main features of the XPS spectra of the Pt, Pd, Pt_80_Pd_20_, and Cu(OH)_2_ NPs. Figure 2a,c show the Pd 3d spectra of the Pd-GP and Pt_80_Pd_20_-GP. The spectrum of Pd-GP was deconvoluted using two doublet components 3d_5/2,3/2_ having 5.3 eV spin-orbit coupling. The spectrum shows a main doublet at 335.5 and 340.8 eV that is attributed to the 3d_5/2,3/2_ of metallic states of palladium (Pd^0^) and a doublet at 337.5 and 342.8 eV that is attributed to Pd(II), indicating a partial surface oxidation [33,34].

The higher binding energy doublet of Pd 3d in the spectrum of the Pt_80_Pd_20_-GP was found at 336.8 and 342.1 eV, respectively, i.e., at 0.7 lower binding energy values. Moreover, the Gaussian relative intensities observed in the Pt_80_Pd_20_-GP system are almost coincident to that of Pd-GP. The latter observation, according to the higher electronegativity of palladium (1.40) than platinum (1.35), could be due to partial charge transfer from platinum to palladium as a consequence of the establishment of a chemical bond between the two metals.

Figure 2b,d show the XPS Pt 4f spectra of the Pt-GP and Pt_80_Pd_20_-GP, respectively. The spectra of Pt 4f were deconvoluted using three doublet components of Pt 4f_5/2,7/2_. The most intense doublet is centered at 71.3 and 74.6 eV of binding energy (3.3 eV spin-orbit coupling) and is assigned to metallic platinum Pt^0^ [35]. The doublet at 72.6 and 76.0 eV is attributed to Pt(II) species. The peaks of the doublet are separated by spin-orbit coupling of 3.4 eV [35,36]. Moreover, the doublet at the higher binding energy values of 75.2 and 78.5 eV (3.3 eV spin-orbit coupling) is assigned to Pt(IV) species [36]. Both binding energy positions and relative intensities of the components of the spectrum of bimetallic NPs are similar to those observed for the Pt-GP sample.

Figure 2e,f show the XPS Cu 2p_3/2_ spectrum and the Cu L_3_M_45_M_45_ Auger peak of the Cu(OH)_2_-GP sample. The Cu 2p peak shows a main envelope centered roughly at about 935 eV and a broad shake up satellite comprised between 942 and 945 eV of binding energy. The main envelope can be deconvoluted by using two Gaussian components centered at 934.9 eV (component A) and 937 eV (component B). The Cu 2p_3/2_ components show an untypical high binding energy with respect to the typical values of binding energy of Cu 2p_3/2_ of copper hydroxide [19]. The observed Cu 2p_3/2_ binding energy can be attributed either to the effect of the particle size as well as to uncompensated Cu(II) cations with a high coordination number of hydroxyl groups, such as Cu(OH)_2_- - OH that possibly represented formally as Cu(III) centers [37,38]. The broad satellite envelope result from a shake-up process occurring in the open 3d^9^ shell, confirming the assignment to the Cu(II) species. The CuL_3_M_45_M_45_ spectrum (Figure 2f) shows a main feature centered at 917 eV of kinetic energy (336.6 eV of binding energy) that confirms the assignments performed on the basis of the binding energy of Cu 2p_3/2_ [19].

### 3.3. Hydrogen Electro-Sorpion, Galvanostatic Polarization, and EIS Characterization

Figure 3a shows the cyclic voltammograms (a third of a total of five cycles) recorded in the potential region ranging between −1 and +0.4 V vs. SCE of the NPs–GP nanoelectrode based on Pt, Pd, Pt_80_Pd_20_ and Cu(OH)_2_. The voltammogram of the GP alone was reported to evaluate the contribution of the support on the voltammograms of NPs. The electro-sorption reaction of hydrogen carried out in alkaline electrolyte shows poor efficiency, compared to that performed in acidic electrolyte. The scientific community agrees that the experimental evidence is due to the slow step of water ionic dissociation into OH^−^ and H^+^ (Equation (1) [38]). Recently, the HER reaction in alkaline media received a renewed interest both in the fundamental research, as well as in industrial application, because of several advantages such as reliability, security, and cheapness [39]. Based on the Volmer–Heyrovsky mechanism, the electro-sorption of hydrogen ions onto the NPs surface can be summarized by the formal Equations (1) and (2) [40].
2H_2_O ⇆ H_3_O^+^ + OH^−^(1)
M + H_3_O^+^ + e^−^ ⇆ MH_2_^+^_ads_ + OH^−^(2)

Juodkazytè and their colleagues propose that the electro-sorption of platinum results in the formation of intermediate molecular hydrogen ions (H_2_^+^) as a precursor to the H_2_ molecule [41]. This assumption is justified because in the acqueous phase water makes the adsorption of H atoms unfavorable [42]. In the anodic direction of the voltammogram of Figure 3a, the peaks A1 and A2 are attributed to the hydrogen oxidation to H^+^ ion and desorption (Equation (3)) and are centered at potentials of −0.68 and −0.49 V (Pt), −0.68 and −0.53 V (Pd), −0.64, and −0.53V (Pt_80_Pd_20_), respectively [20]. The two anodic peaks present in the voltammograms are attributable to the oxidation mechanism of MH_2_^+^_ads_ to M + 2H_3_O^+^ on different crystallographic planes of the NPs’ surface [43]. According to this explanation, the voltammogram of GP alone shows a single peak centered at −0.31 V.
MH_2_^+^_ads_ + 2H_2_O ⇆ M + 2H_3_O^+^ + e^−^
(3)

The peak A3 at −0.48 V is attributable to the electro-sorption reaction of water onto copper that, in turn, is produced by the reduction of Cu(OH)_2_ during the cathodic sweep of the potential down to −1 V. The reaction that originates the peak A3 is summarized by the Equation (4).
Cu + 2H_2_O ⇆ Cu(OH)_ads_ + H_3_O^+^ + e^−^(4)

The peak A4, located at −0.25 V, is attributable to the electro-sorption of water onto Cu(OH)_ads_ and formation of a complex hydrous of Cu(II) specie [44,45].
Cu(OH)_ads_ + 2H_2_O ⇆ [Cu(OH)_2_]_ads_ + H_3_O^+^ + e^−^(5)

Moreover, additional peaks A5, located at about −0.11 V, are observed in the voltammograms of Pt, Pd, and Pt_80_Pd_20_ NPs that can be attributed to the hydrogen spillover process [20,46]. Based on this mechanism, the adsorbed hydrogen onto the NPs passes to the GP, (Equation (6)).
MH_2_^+^_ads_ + GP + H_2_O ⇆ M + GPH_ads_ + H_3_O^+^(6)

It is worthy of note that Cu(OH)_2_ NPs does not show the peak A5 because the electro-sorption of H onto this material is not favored (vide Equations (4) and (5) and Table 1). In the cathodic direction of the voltammograms, the peaks C1 are assigned to the adsorption and reduction of the H_3_O^+^ ions onto the NPs’ surface (Equation (2)). The peaks C1 are observed at the potentials of −0.38 V (GP alone), −0.37 V (Pt), −0.43 V (Pd), and −0.35 V (Pt_80_Pd_20_), respectively. The Pt_80_Pd_20_ alloy shows the lowest potential, according to its greater efficiency of electro-catalysis toward the hydrogen electro-sorption process.

The Butler–Volmer Equation [47] suggests that the shifts of the cathodic peak toward higher potential values are associated with the increase in its full width at half maximum (FWHM). Based on this consideration, the peak C1 of Pt_80_Pd_20_ is characterized by a larger FWHM than the peaks obtained by the Pt or Pd.

**Table 1 nanomaterials-13-00561-t001:** Adsorption energies of H, H_2_O, and OH species onto Cu(111), Pt, and Pd surfaces.

Adsorbed Species	Adsorption Energies (eV)				Reference
	Cu(111)	Cu(110)	Pt(001)	Pd(111)	
H_2_O	0.12	−0.43 ^(4,5)^	–1.67	−1.5 ^(3)^	[48,49,50,51]
H·	2.57	-	0.11 ^(2)^	0.98 ^(4)^	[42,49,52]
H_2_O	0.28 ^(1)^	-	-	-	[49]
OH·	2.04 ^(1)^	-	-	-	[49,50]

^(1)^ Oxygen covered Cu(111) surface; ^(2,3)^ extrapolated from the reference [49]; ^(4)^ experimental data; ^(5)^ extracted from kinetic data in Nakamura et al. [48].

Figure 3b reports the voltammograms to an enlarged scale in the potential range between −0.5 to −1 V. Cu(OH)_2_ NPs in the cathodic direction shows two peaks (C2 and C4) at −0.53 and −0.78 V, respectively, that are assigned to the reduction of Cu(II) to Cu(I) and Cu(I) to Cu°, respectively [45]. Moreover, a substantial rise in cathodic current is seen at a potential of -0.9 V (C3), which can be attributed to the hydrogen evolution reaction on Cu(OH)_2_, per Equations (7) and (8):Cu(OH)_2_ + H_2_O ⇆ Cu(OH)_2_ - - (HOH)_ads_(7)
Cu(OH)_2_ - - (HOH)_ads_ + e^−^ ⇆ [Cu(OH)_3_]^−^ + ½ H_2_(8)

In accordance with Dias Martins and coworkers [10], we propose the formation of an activated complex Cu(OH)_2_ - - (HOH)_ads_ between water and highly hydrophilic nanowalls of Cu(OH)_2_ prepared by PLAL reduces the energy barrier for water dissociation, which produces a significant activity of HER in alkaline environments.

The mechanisms proposed by us through the Equations (1)–(5) are supported by the adsorption energies of H, H_2_O, and OH onto CuOx, Pt, and Pd surfaces, respectively. Table 1 summarizes the energies taken from the literature for the systems here considered. The adsorption energy of H, shown in Table 1, is more endothermic on Cu than on Pt and Pd. These absorption energies explain the well-known poor properties Cu as a catalyst toward HER, compared to Pt or Pd. Exothermic values are shown for water adsorption on Pt and Pd. Data derived from Nakamura [48] reveal an exothermic value for water adsorption also on the surface of Cu(110). Moreover, water adsorption energy is more exothermic on Pt and Pd than on Cu or CuOx. The adsorption energy of OH onto a copper surface is significantly endothermic; therefore, a mechanism involving its direct adsorption is not likely. The proposed complex Cu(OH)_2_ - - (HOH)_ads_ (Equations (7) and (8)) on the highly hydrophilic surface of Cu(OH)_2_ is expected to reduce the barrier energy for water adsorption and dissociation, based on a mechanism similar to that proposed by Dias Martins and coworkers [10].

Figure 3c,d show the voltammograms of the NPs drop cast by the dispersion of water 0.25 wt.% of Nafion. The voltammogram shows the same peaks of Figure 3a, but they are less pronounced. This finding can be explained through the increased and finely dispersion of the NPs in this sample that work as nanoelectrode arrays (vide Figure 1) [53]. It is notable that the C3 feature is not observed, as reactions 7 and 8 are limited by the presence of Nafion.

Galvanostatic charge and discharge measurements were performed to further characterize the electrochemical properties of the systems. Figure 4a shows the curves of galvanostatic charge and discharge obtained by the NPs–GP systems. For comparison, the curves obtained by the GP alone were reported. The potential of the electrode was reported as the function of the specific capacity. The fifth cycle of charge–discharge processes has been reported. At this cycle, we observed the stabilization of the electrode potential both in the charge as well as in the discharge direction. In our experiment, the electrodes were charged at a constant current of −100 μA up to a capacity of 7.8 Ahg*^−^*^1^ and then discharged at constant current of +100 μA. The specific capacity was calculated according to Faraday’s law and the works reported by other authors [54,55,56]. In our experiments, at a capacity of 7.8 Ahg*^−^*^1^, the potentials of both charge and discharge processes reach steady state values. At this stage, the equilibria of the most relevant electrochemical processes schematized through the Equations (2)–(8) are established. The electrode potential of Pt-GP (Figure 4a) increases faster than that of the GP alone at the beginning of both charge and discharge processes. This finding reflects the high efficiency of this electro-catalyst, particularly in the form of nanoparticles. The Pd-GP and Pt_8_Pd_20_-GP show a higher rate of increase in the charge and discharge electrode potential than Pt, which reflects their superior efficiency with respect to Pt in the HER process, especially when Pd is co-catalyst with Pt [16]. The Cu(OH)_2_ shows even more negative electrode potential in the charge curve than the noble metals, indicating its significant efficiency toward HER. In the discharge curve, the electrode potential of the Cu(OH)_2_ is less than of the noble metals, according to the different mechanisms of electro-sorption and desorption reactions onto this materials, previous discussed.

The total amount of hydrogen developed, stored in the NPs or spilled-out to the GP, corresponding to the steady state potential of polarization curves, was reported in Table 2 (the method of the calculation is given later on). From these methods, we obtained that the total amount of hydrogen developed is equal to 1 wt.% for the platinum, 6 wt.% both for the palladium, and the bimetallic Pt_80_Pd_20_ and 2 wt.% for Cu(OH)_2_.

Figure 4b shows the curves of galvanostatic charge and discharge of the NPs–GP drop cast by water-0.25 wt.% of Nafion suspension. According to the Equation (9), the Nafion layer that surrounds the NPs surface acts as thin proton-exchange membranes. Because the Nafion is H^+^ permeable, the hydrogen ions activity on the NPs surfaces increases (Equation (10)):(CF_n_O_m_)_q_-SO_3_^−^ + H_2_O ⇆ (CF_n_O_m_)_q_-SO_3_^-^ - H^+^ + OH^-^(9)
(CF_n_O_m_)q-SO_3_^−^ - H^+^ + NPs + e^−^ ⇆ NPsH_ads_ + (CF_n_O_m_)_q_-SO_3_(10)
where (CF_n_O_m_)_q_-SO_3_^−^ and (CF_n_O_m_)_q_-SO_3_^−^ - H^+^ represent the ionomer in the anionic and H^+^ bound forms, respectively. The charging curves are very close to each other and reach almost the same potential values for all the NPs. This finding indicates that the limiting steps of the reaction is the same in all the systems and are determined by the kinetics of the Reactions (9) and (10). Conversely, the trend for the discharge curves potential at steady state is similar to that observed for NPs without Nafion, with the exception of Pd NPs. The findings maybe explained if we assume that the desorption of hydrogen from the NPs surface, once it has been electro-adsorbed, is not influenced primarily by the presence of the Nafion layer. 

EIS characterization was performed to further analyze the behavior of the NPs. Figure 5a reports the Nyquist plot of the imaginary impedance component as function of real impedance component. The Cu(OH)_2_ shows the lowest value of real impedance, among the considered nanomaterials, which reflects the low charge transfer resistance toward HER. Figure 5b reports the corresponding Bode plot of impedance (modulus) that was measured in the frequency range of 1 to 10^4^ Hz. Figure 5c shows the Bode plot for the impedance phase. The lowest modulus of impedance is exhibited by the Pt NPs. For purposes of clarity, Figure 5d shows a histogram of the phase and impedance modulus for the various systems measured at a frequency of 1 Hz. Noteworthy is that the modulus and phase of the Cu(OH)_2_ NPs are closest to that of Pt NPs. Moreover, the phase of impedance, at the lowest frequency of 1 Hz, ranges from 52 to 62 degrees (Figure 5d, blue bars). The capacitive characteristics of the interface between the electrode and solution can be considered similar in all the systems here considered. Therefore, the observed differences on the polarization curves of Figure 4a,b can be attributed mainly to different rates of the faradaic processes. 

Table 2 summarizes the performance of our nanomaterials in comparison to that of nanomaterials based on Pt, Pd, Ni, Cu, and Ag, developed for HER and/or hydrogen storage, as described in the literature [20,56,57,58,59,60,61,62,63]. The nanomaterials were obtained by conventional methods, by PLAL, or by laser ablation. Some systems were obtained by treating the surfaces of Cu and Ni sheets with femtosecond laser ablation [57,58]. Because the literature data regarding electro-sorption and hydrogen storage are often inhomogeneous, we have reported only those parameters that can be more easily compared. The curves in Figure 4a,b allow us to determine the charge capacity of our systems, as previously discussed. For this purpose, we assumed the start of the charging process when the electrode potential reached the value of −0.3 V with respect to SCE. This value corresponds to 77% (Cu(OH)_2_) to 91% (Cu(OH)_2_ – Nafion) of the steady state potentials of the various NPs based electrodes. Instead, in the discharge process we have chosen the cut-off potential at +0.25 V with respect to SCE, which corresponds approximately from 80% (for noble metals) to 100% (for Cu(OH)_2_) of the potential assumed by the electrode at the steady state. This procedure was based on methods reported in previous works which reported measurement methodologies similar to ours [52,54,60]. The faradaic efficiency was calculated from the percentage of the ratio of the discharge capacity to the charge capacity, according to previous works [62]. The highest faradaic efficiency we obtained is shown by the Pt-Nafion system with a value of 86.6% [20]. This result is related to the increase in dispersion of platinum NPs by Nafion, according to the morphology shown in Figure 1e. It should be noted that the highest charge and discharge capacity values are presented by Pd and Pt_80_Pd_20_, with a faradaic efficiency of about 78% corresponding to the bimetallic system. When Nafion is present, the highest charge capacity is, however, shown by the bimetallic system with a faradaic efficiency of about 72%. Taking into account the surface concentration of platinum in Pt-GP obtained by XPS characterization, which is 18 times higher than that of palladium in Pd-GP and 9 times higher than that of the sum of palladium and platinum in the bimetallic system [20], the highest charge–discharge capability is exhibited by systems containing palladium. The interaction between the two metals with consequent electron donation from platinum to palladium can explain the high electro-sorption capacity of H^+^ ions and, therefore, the charge capacity of the bimetallic system. Our data are in agreement with the literature [16], confirming the superior performance of the palladium-platinum-based co-catalyst compared to the platinum-based ones toward the H^+^ electro-sorption reaction. The Cu(OH)_2_ system obtained by PLAL has good HER efficiency, but the faradaic efficiency is very low, according to the properties of the material which is unsuitable for hydrogen storage. According to Lao [57] and Poimenidis [58], the laser treatment at femtosecond produces a significant improvement of the HER efficiency of copper and nickel sheets, respectively. The fifth column of Table 2 shows the absorption data at a pressure of 1 bar and at room temperature, obtained by chemical reaction in the gaseous phase or by electro-sorption reaction, respectively. The data comparison shows that our Cu(OH)_2_ system prepared by PLAL is competitive compared to other state-of-the-art nanomaterials. The proposed synthesis method is low cost and involves a very low quantity of non-critical raw material of the order of 0.1 – 1 mg m^−2^ of graphene paper. Moreover, the synthesis process has a low environmental impact because it does not lead to the formation of harmful chemical waste.

**Table 2 nanomaterials-13-00561-t002:** Evolution and adsorption of hydrogen by the Pt, Pd, PtPd, Cu(OH)_2_, Pt -Nafion, Pd -Nafion, PtPd -Nafion, and Cu(OH)_2_ -Nafion, in comparison to platinum, copper, nickel, and palladium nanoparticles-decorated carbon nanomaterials reported in the literature.

NPs	Electrode	NPs Production Method	Electrolyte or Gaseous Phase Reaction of H_2_ Adsorption	Hydrogen Storage/Evolution (wt.%)	Faradaic Efficiency (%)	Reference
Cu foil untreated	Cu	-	KOH 1 M	159 ^(3)^		[57]
Cu foil black Nanostructured surface	Cu	Laser 800 nm, pulse width 35 fs	KOH 1 M	1 ^(3)^	-	[57]
Ni foil with nanostructured surface	Ni	multipass Ti: Sapphire laser amplifier pulse width 30 fs	KOH 1 M	10 ^(4)^		[58]
Ni/rGO	-	Reduction in H_2_ ^(1)^	Gaseous phase reaction	0.007	-	[59]
Ni/Pd/rGO,	-	Reduction in H_2_ ^(1)^	Gaseous phase reaction	0.13	-	[59]
Ni/Ag/Pd/rGO	-	Reduction in H_2_ ^(1)^	Gaseous phase reaction	0.055	-	[59]
Pd-Nafion	GCE	Wet/NaBH_4_	H_2_SO_4_ 0.5 M	0.003	83.1	[56]
Pd-rGO/Nafion	GCE	Wet/NaBH_4_	H_2_SO_4_ 0.5 M	0.14	85	[56]
Pd/B-rGO/Nafion	GCE	Wet/NaBH_4_	H_2_SO_4_ 0.5 M	0.35	95	[56]
Pt Covalent triazine framework (CTF-N)	Fluorine doped tin oxide (FTO)	Wet/NaBH_4_	Trietanolamine ^(2)^	0.2	-	[61]
Pd Covalent triazine framework (CTF-N)	Fluorine doped tin oxide (FTO)	Wet/NaBH_4_	Trietanolamine ^(2)^	1.05	-	[61]
Pt- (GO)/HKUST-1	-	Wet/Ethylene glycol	Gaseous phase reaction	1.6	-	[62]
Pd/graphene	-	Reduction in H_2_ ^(1)^	Gaseous phase reaction	8.67 ^(5)^	-	[63]
Pd/MWCNT	-	PLAL	Gaseous phase reaction	1.2	-	[60]
Pt	GP	PLAL	KOH 1 M	1	64.2	[20]
Pd	GP	PLAL	KOH 1 M	6	66.1	[20]
Pt_80_Pd_20_	GP	PLAL	KOH 1 M	6	77.9	[20]
Pt-GP-Nafion	GP	PLAL	KOH 1 M	1	86.6	[20]
Pd-GP-Nafion	GP	PLAL	KOH 1 M	6	47.8	[20]
Pt_80_Pd_20_-GP-Nafion	GP	PLAL	KOH 1M	6	72.4	[20]
Cu(OH)_2_	GP	PLAL	KOK 1M	2	17	This work
Cu(OH)_2_-GP-Nafion	GP	PLAL	KOH 1M	2	-	This work

^(1)^ reduction at 300 °C, H_2_ atmosphere; ^(2)^ photoelectrochemical method; ^(3)^ charge transfer resistance of the electrode (ohm cm^2^); ^(4)^ hydrogen developed as gas phase; ^(5)^ value referred to a pressure of 60 bar, otherwise unspecified values were measured at 1 bar.

## 4. Conclusions

Unconventional synthesis of nanomaterials of Pt, Pd, Pt_80_Pd_20_, and Cu(OH)_2_ by PLAL produces NPs that have high activity as an electro-catalyst toward the hydrogen electro-sorption in alkaline media. Noble-metal NPs show a typical spherical shape, according to their melting during the synthesis process. They are mainly formed by the unoxidized metals, in accordance with their known poor reactivity with water used in the PLAL process. Cu target under PLAL produce the formation of Cu(OH)_2_ nanostructure having the flower-bud-like shape, as a result of the chemical reaction with water. Moreover, the nanostructures are characterized by very thin nanowalls of Cu(OH)_2_. The XPS spectra of Cu(OH)_2_ are compatible with uncompensated centers of Cu(II), characterized by a high co-ordination number with OH^−^ and H_2_O species. According to similar intermediate proposed in the literature, we propose the formation of an intermediate with water Cu(OH)_2_ - - HOH_ads_ can reduce the barrier energy of water adsorption and dissociation. This mechanism leads to the increase of the rate of the HER in alkaline media. The presence of the ionomer promotes the NPs dispersion onto the surface of GP and is expected to increase the ionic activity of H^+^ close to the NPs surface and, after, the rate of the HER.

## Figures and Tables

**Figure 1 nanomaterials-13-00561-f001:**
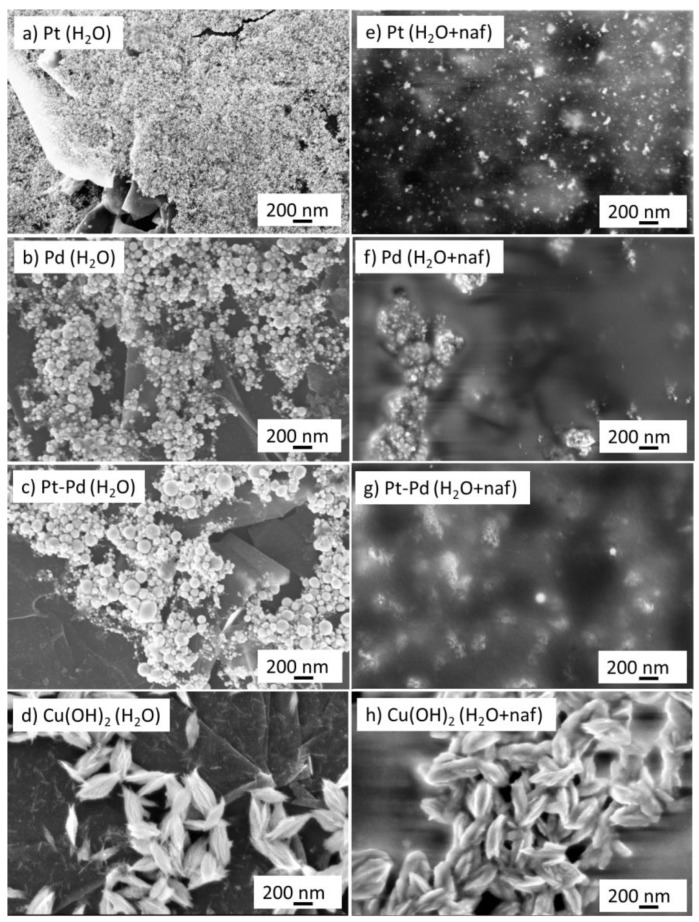
Field emission scanning electron micrographs of NPs: (**a**) Pt; (**e**) Pt-Nafion; (**b**) Pd; (**f**) Pd-Nafion; (**c**) Pt_80_Pd_20_; (**g**) Pt_80_Pd_20_-Nafion; (**d**) Cu(OH)_2_; (**h**) Cu(OH)_2_-Nafion.

**Figure 2 nanomaterials-13-00561-f002:**
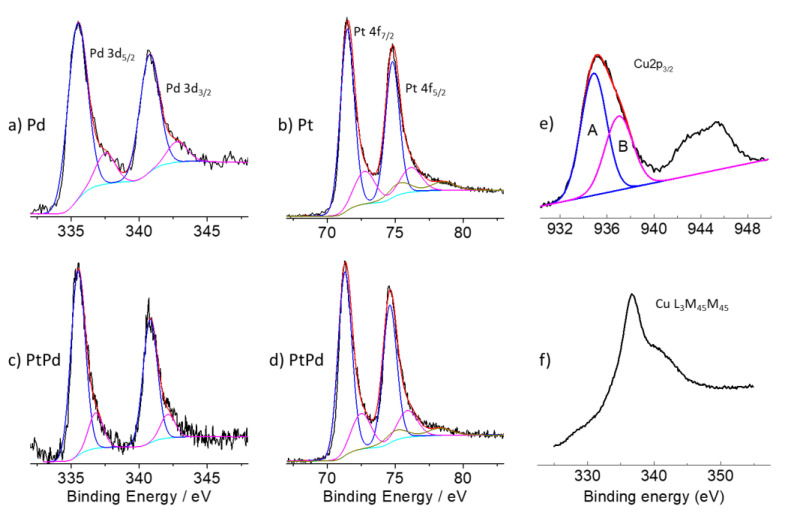
Photoelectron spectra of the regions: (**a**,**c**) Pd 3d of Pd-GP, and Pt_80_Pd_20_-GP, respectively. The 3d_5/2_—3d_3/2_ spin-orbit doublets (blue and magenta line) refer to the Pd^0^ and Pd(II) states, respectively; (**b**,**d**) Pt 4f of Pt-GP, and Pt_80_Pd_20_-GP, respectively. The 4f_7/2_—4f_5/2_ spin-orbit doublets (blue, magenta, and dark yellow line) refer to Pt^0^, Pt(II) and Pt(IV) states, respectively. The cyan line refers to the background and the red line superimposed to the experimental data, referring to the sum of all of the Gaussian components; (**e**,**f**) Cu 2p _3/2_ and Cu L_3_M_45_M_45_ Auger peak of Cu(OH)_2_.

**Figure 3 nanomaterials-13-00561-f003:**
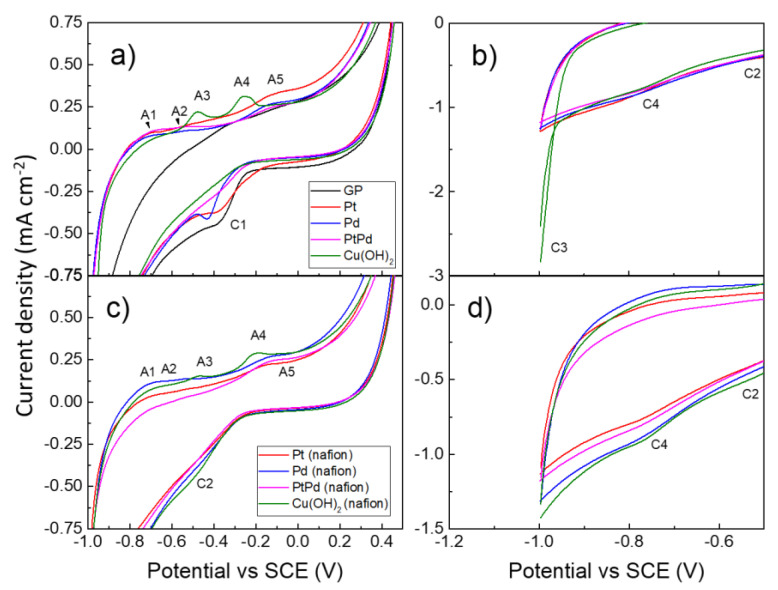
Cyclic voltammograms of GP alone (black line) Pt (red line), Pd (blue line), Pt_80_Pd_20_ (magenta line), and Cu(OH_2_ (olive line); (**a**,**b**) drop casting by water suspension; (**c**,**d**) drop casting by suspension in water 0.25% wt. Nafion. Conditions: KOH 1 M; scan rate 20 mVs*^−^*^1^.

**Figure 4 nanomaterials-13-00561-f004:**
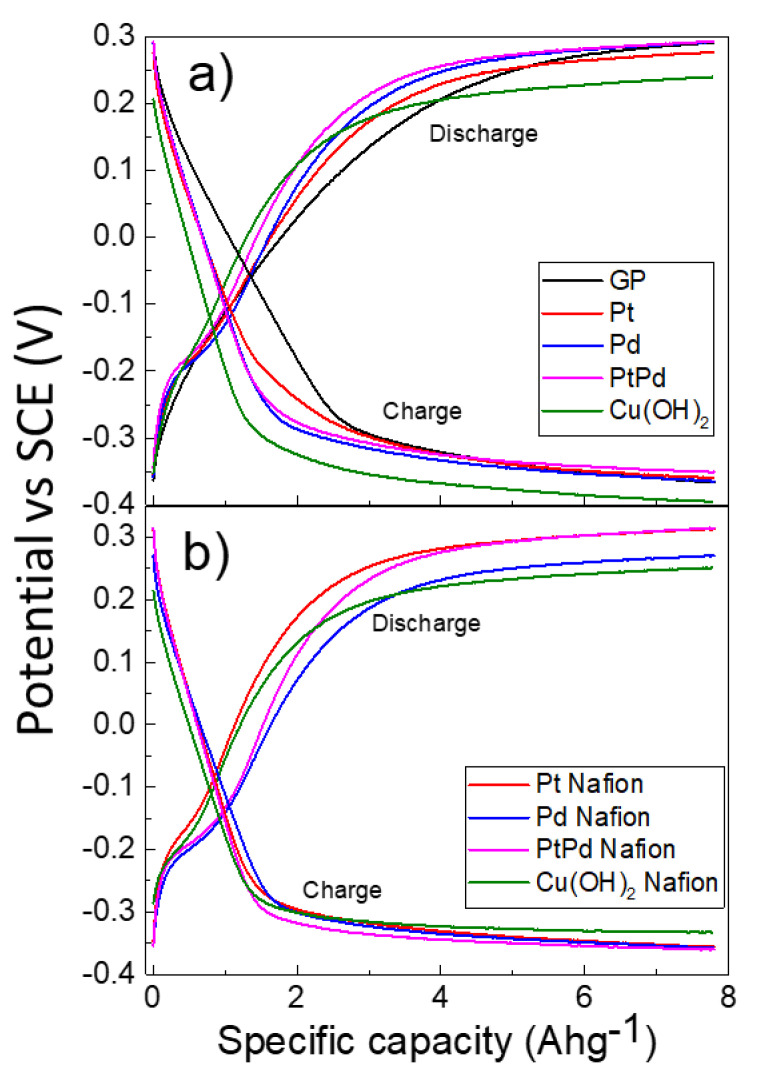
Galvanostatic charge and discharge curves of GP alone (black line), Pt (red line), Pd (blue line), Pt_80_Pd_20_ (magenta line), and Cu(OH_2_ (olive line): (**a**) drop casting by water suspension; (**b**) drop casting by suspension in water 0.25% wt. Nafion. Conditions: KOH 1 M; current ±100 μA. The specific capacity was calculated with respect to the mass of 1 cm^2^ of GP.

**Figure 5 nanomaterials-13-00561-f005:**
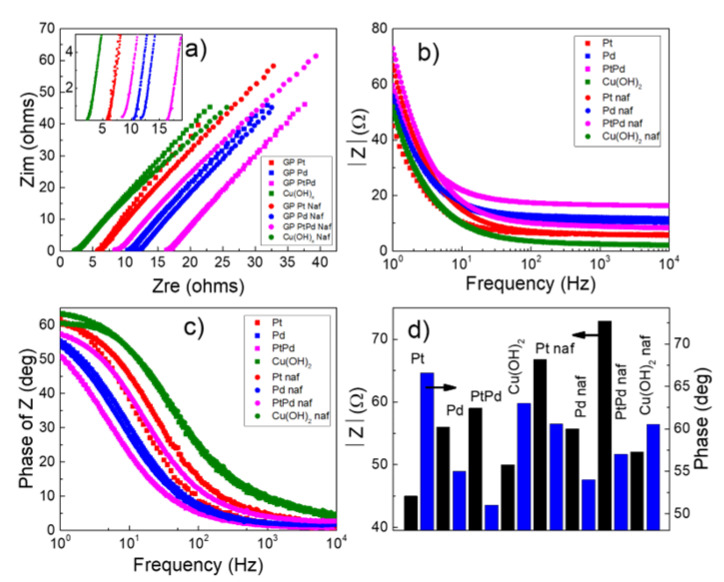
Electrochemical Impedance Spectroscopy characterization of Pt (square red data), Pd (square blue data), Pt_80_Pd_20_ (square magenta data), Cu(OH)_2_ (square olive data), Pt-Nafion (circle red data), Pd -Nafion (circle blue data), Pt_80_Pd_20_ -Nafion (circle magenta data), and Cu(OH)_2_ -Nafion (circle olive data) NPs: (**a**) Nyquist plot of the imaginary impedance as function of the real impedance. The inset shows the enlarged region of the spectrum of low impedance; (**b**) modulus of impedance as function of frequency; (**c**) corresponding phase spectra; (**d**) impedance modulus |Z| (black bars) and phase of Z (blue bars) measured at frequency of 1 Hz. Condition: KOH 1 M; current density: 250 μA cm^−2^ rms.

## Data Availability

The detailed data of the study are available from the corresponding authors by request.

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
