# Peer review of "Alkaline Electro-Sorption of Hydrogen Onto Nanoparticles of Pt, Pd, Pt80Pd20 and Cu(OH)2 Obtained by Pulsed Laser Ablation"

_nanomaterials, 2023, doi:10.3390/nano13030561_

Round 1
Reviewer 1 Report
In this work, the authors investigate the feasibility of the HER process in an alkaline medium on catalytic electrodes formed on graphene paper and coated with nanoparticles of various metals obtained in a pure water medium by the PLAL method. They consider this to be a very simple and effective method for the preparation of such particles. The authors examine in detail the electrochemical properties of the electrodes obtained, compare them with literature data and conclude that, in the alkaline medium, the efficiency of the HER process is determined by the facilitation of the dissociation process of water molecules and, at the same time, by the increase in the adsorption of all H-containing particles on the surface of the electrode. The paper is clearly written, the methodology used is state-of-the-art and the results obtained are unquestionable.
The paper can be accepted for publication. Several minor points to consider:
1) it is not necessary to write "1e-" in reaction equations, just "e-" is enough.
2) the authors should specify what is meant by H and OH in Table 1. H atom and OH radical?
3) a recent paper which might be of interest for the authors DOI: 10.3390/catal11091135
Author Response
Reviewer #1
In this work, the authors investigate the feasibility of the HER process in an alkaline medium on catalytic electrodes formed on graphene paper and coated with nanoparticles of various metals obtained in a pure water medium by the PLAL method. They consider this to be a very simple and effective method for the preparation of such particles. The authors examine in detail the electrochemical properties of the electrodes obtained, compare them with literature data and conclude that, in the alkaline medium, the efficiency of the HER process is determined by the facilitation of the dissociation process of water molecules and, at the same time, by the increase in the adsorption of all H-containing particles on the surface of the electrode. The paper is clearly written, the methodology used is state-of-the-art and the results obtained are unquestionable.
Author reply: we would like to thank this Reviewer for his/her very competent and unbiased review process.
The paper can be accepted for publication. Several minor points to consider:
1) it is not necessary to write "1e-" in reaction equations, just "e-" is enough.
Author reply: thank you for this suggestion, we deleted “1” in all the equations.
2) the authors should specify what is meant by H and OH in Table 1. H atom and OH radical?
Author reply: Yes, we considered them in the form of radical, we added the symbol of radical “·” in Table 1, thank you for the tips.
3) a recent paper which might be of interest for the authors DOI: 10.3390/catal11091135
Author reply: Thank you very much for this reference. I studied it very careful and added this reference at line 226, as new Ref 41. It is a good paper. The reference pointed out on some mechanism of HER on Pt and on the fact that the HER reaction is the same both in the acidic and in alkaline pH. The paper suggests H2+ads as intermediate in the electro-sorption of hydrogen on noble metals. We used this intermediate through the equations 2-5.

Reviewer 2 Report
1. Line 99 and 100"The electrodes were prepared by drop casting 29 μg cm−2 of platinum, or 99 5 μg cm−2 of palladium, or 5 μg cm−2 of Pt80Pd20 or 15 μg cm−2 of Cu(OH)2." Why different amounts but not consistent?
2. line 118 "In each measurement, 30 mL of fresh solution of KOH 1 M was used." Why 1M KOH but not lower concentration?
3. Line 131 and 132 "The structures of these NPs were studied by XRD 131 that revealed their crystalline nature [20]. The NPs have an average size of 8 to 16 nm [20]" Is this your own measurement or just cited?
4. Line 159-161 " The latter observation, according to the larger electronegativity of palla- 159 dium (1.40) with respect to that of platinum (1.35), can be explained with the establish- 160 ment of a chemical bond between the two metals, that results in electron donation from 161 platinum to palladium". How did you confirm this statement?
5. In figure 5a, can the frequency (low Ohm) region be zoomed in for at least one data?
6. Line 341 and 342, "The 341 equivalent RC circuits can be considered similar in all the system here considered". I could not see the equivalent circuit in the figure or anywhere in the paper.
7. Line 352-354, "For this purpose, we assumed the 352 start of the charging process when the electrode potential reached the value of -0.3 V vs 353 SCE." How did you convert the potential to SCE reference value?
8. Line 358 and 359, "For this purpose, we assumed the 352 start of the charging process when the electrode potential reached the value of -0.3 V vs 353 SCE." what is the reference potential? Is it also -0.3V?
9. Line 362 and 363, "Palladium and Pt80Pd20 show the highest charge 362 and discharge capacity values," At what potential is it?

Author Response
Reviewer #2
Author reply: we would like to thank this Reviewer for his/her very competent and unbiased review process.
- Line 99 and 100"The electrodes were prepared by drop casting 29 μg cm−2 of platinum, or 5 μg cm−2 of palladium, or 5 μg cm−2 of Pt80Pd20 or 15 μg cm−2 of Cu(OH)2." Why different amounts but not consistent?
Author reply: thank you for this question. The reason is that the ablation rate is different for the various materials, then, using the same laser conditions and ablation time, the amount of ablated NPs in the ware dispersion is different. We prefer do not dilute or concentrate the dispersions to obtain the same amount of NPs, to avoid some possible and unknown modification of the stability of the NPs dispersions.
- line 118 "In each measurement, 30 mL of fresh solution of KOH 1 M was used." Why 1M KOH but not lower concentration?
Author reply: 1 M is the typical concentration of KOH reported in literature, so we have chosen this value to have a comparison with the literature.
- Line 131 and 132 "The structures of these NPs were studied by XRD 131 that revealed their crystalline nature [20]. The NPs have an average size of 8 to 16 nm [20]" Is this your own measurement or just cited?
Author reply: the Ref [20] is our own previous publication, so the data are of our team [please, see also ref. 32].
- Line 159-161 " The latter observation, according to the larger electronegativity of palladium (1.40) with respect to that of platinum (1.35), can be explained with the establishment of a chemical bond between the two metals, that results in electron donation from platinum to palladium". How did you confirm this statement?
Author reply: this is just a hypothesis, and in the revised version we used a conditional expression to describe this point. The reason is that in the alloy the BE of the Pd3d assigned to Pd(II) is 0.7 eV lower with respect to the Pd alone. Roughly, in the “frozen orbital approximation” the difference in the electronegativity of two bonded atoms represents a cause of binding energy shift.
- In figure 5a, can the frequency (low Ohm) region be zoomed in for at least one data?
Author reply: to cope with this reviewer’s suggestion, we add an inset in Figure 5a that shows the low impedances region of the spectrum.
- Line 341 and 342, "The equivalent RC circuits can be considered similar in all the system here considered". I could not see the equivalent circuit in the figure or anywhere in the paper.
Author reply: We have not included the simulation with the electrical equivalent circuit because out of the scope of our work. In fact, our work is mainly focused on the properties of NPs produced by PLAL towards the HER. The simulation with electrical equivalent circuit of the electrode it would have rendered the article out of purpose for which it was intended. Even if we have not included any equivalent circuit scheme, the electrode-solution interface can be simulated by series – parallel RC network. Basing on our results of Figure 5a-d, we concluded that the capacitive contribution is similar in all the systems because the phase and impedance modulus (Figure 5d) are not very different in the various systems analyzed.
- Line 352-354, "For this purpose, we assumed the start of the charging process when the electrode potential reached the value of -0.3 V vs SCE." How did you convert the potential to SCE reference value?
Author reply: we did not converted the potential to SCE, but the potential was measured experimentally during the galvanostatic charge and discharge curves. The electrode potential is measures vs. SCE.
- Line 358 and 359, "For this purpose, we assumed the start of the charging process when the electrode potential reached the value of -0.3 V vs SCE." what is the reference potential? Is it also -0.3V?
Author reply: the reference potential is that of SCE, i.e. +0.248 V vs. SHE at 20 °C and +0.244 V vs. SHE at 25 °C. Similar measurements were done by:
Boateng, E.; Dondapati, J.S.; Thiruppathi, A.R.; Chen, A. Significant enhancement of the electrochemical hydrogen uptake of reduced graphene oxide via boron-doping and decoration with Pd nanoparticles. Int. J. Hydrog. Energy 2020, 45, 28951–28963, but using the Ag/AgCl instead of SCE reference electrode (please vide the attached Figure).
To cope with this reviewer’s comment, we clarified this point better at the lines 139-141 of the revised manuscript.
- Line 362 and 363, "Palladium and Pt80Pd20 show the highest charge and discharge capacity values," At what potential is it?
Author reply: thank you for question. Charge and discharge specific capacity of the NPs-GP systems have been calculated at potential of -0.3 V and +0.25 V vs. SCE, respectively. Faradaic efficiency was calculated by the discharge to charge capacity percentage ratio. We add new phrase at lines 146-148 of the experimental section to clarify this point.

Reviewer 3 Report
Manuscript number: nanomaterials-2177131
Title: Alkaline electro-sorption of hydrogen onto nanoparticles of Pt, Pd, Pt80Pd20 and Cu(OH)2 obtained by Pulsed Laser Ablation
1/ The paper contains some grammatical errors and typo-mistakes that should be corrected.
2/ Abbreviations should be opened when mentioned for the first time.
3/ The Abstract part should be improved. Shortened the background in the Abstract and the authors should clearly inform the important findings in the present study within the Abstract.
4/ The abstract should contain some qualitative and quantitative results.
5/ The authors should clarify the choice of Graphene paper (GP).
6/ Details and parameters of PLAL should be provided in the current manuscript.
7/ SEM images in Figure 1 are already in the previous publication of the author (https://doi.org/10.3390/mi13060963). The authors should provide other images to avoid duplication.
8/ Similarly, the XPS spectra presented in Figure 2 are the same presented in a previous publication made by the authors (https://doi.org/10.3390/mi13060963).
9/ Similarly, the results of GP, Pt, Pd, and PtPd presented in Figures 3 and 4 are already published in the previous study made by the authors (https://doi.org/10.3390/mi13060963).
10/ I would recommend the authors to present the results related to Cu(OH)2 and just discuss/compare the results of Pt, Pd, Pt80Pd20 by referencing to them.
Due to these difficulties (duplication of results), I can NOT recommend of the present study.
Author Response
Reviewer #3
Title: Alkaline electro-sorption of hydrogen onto nanoparticles of Pt, Pd, Pt80Pd20 and Cu(OH)2 obtained by Pulsed Laser Ablation
Author reply: we would like to thank this Reviewer for his/her very competent and unbiased review process.
1/ The paper contains some grammatical errors and typo-mistakes that should be corrected.
Author reply: we carefully checked and corrected the grammatical errors, thank you.
2/ Abbreviations should be opened when mentioned for the first time.
Author reply: we corrected this error when occurred.
3/ The Abstract part should be improved. Shortened the background in the Abstract and the authors should clearly inform the important findings in the present study within the Abstract.
Author reply: according to the reviewer comment, the abstract has been improved and shortened.
4/ The abstract should contain some qualitative and quantitative results.
Author reply: we improved the abstract by adding new results at new lines 30, 35 and 36.
5/ The authors should clarify the choice of Graphene paper (GP).
Author reply: to cope with this reviewer’s comment, we add a new sentence at lines 90-93.
6/ Details and parameters of PLAL should be provided in the current manuscript.
Author reply: to cope with this reviewer’s comment, we add new sentences at lines 98-101 that report the experimental conditions used in the PLAL experiments.
7/ SEM images in Figure 1 are already in the previous publication of the author (https://doi.org/10.3390/mi13060963). The authors should provide other images to avoid duplication.
Author reply: in order to help the readers we have reported and compared the data of the previous work with the new ones relating to Cu(OH)2. We have reported in several places of the manuscript the reference to our previous work. In our opinion, the Figures, while reporting some experimental data that have already been published, should in any case be considered new as they contain the new data relating to the Cu(OH)2. Furthermore, some reviewers advised us to report the data from the previous work, e.g. details on the experimental parameters for the synthesis of the NPs, just to help the readers and not force them to cross-read the various publications.
8/ Similarly, the XPS spectra presented in Figure 2 are the same presented in a previous publication made by the authors (https://doi.org/10.3390/mi13060963).
Author reply: please see author reply to the point 7.
9/ Similarly, the results of GP, Pt, Pd, and PtPd presented in Figures 3 and 4 are already published in the previous study made by the authors (https://doi.org/10.3390/mi13060963).
Author reply: please see author reply to the point 7.
10/ I would recommend the authors to present the results related to Cu(OH)2 and just discuss/compare the results of Pt, Pd, Pt80Pd20 by referencing to them.
Author reply: please see author reply to the point 7.
Due to these difficulties (duplication of results), I can NOT recommend of the present study.
The reference https://doi.org/10.3390/mi13060963 that is our own publication in Micromachines and the present manuscript are completely different in several part and contain different Figures. Just some Figures contain the same data concerning the noble metal NPs to help the readers in the comparison of them with the behavior and characteristics of Cu(OH)2 in the same plot or FE-SEM pictures. The objective was to avoid the readers from inter-reading between two different articles. The actual manuscript contains the data of Cu(OH)2 and, then, it is to be considered new. Moreover, based on the specific comments and on the weak points identified by all the four Reviewers, we deeply revised our manuscript in several parts and improved the literature references, which are now stronger and more solid. In particular, we rewrote the Equations 2-5 using the specie H2+ads instead of Hads. This new intermediate is based on recent literature results reported in the new references [26 and 41]. For these new aspects, the paper has to be considered original in various parts and not a duplication. We thus believe that the manuscript is now acceptable for publication.

Reviewer 4 Report
The manuscript is of interest for the journal but it needs revision prior to being considered for acceptance. After revision, it can be reconsidered. Below are comments and suggestions to help the authors improve it:
1\ English should be improved here and there, for example, in Conclusions: (a) synthesis ... produce ; (b) an high ; (c) produces the formation. Also, in Abstract, line 20: probably should read "consists in reducing..."
2\ Line 448: must be Electrochem. Commun. (not commun.)
3\ Line 97: instead of "drop casted" ,it should be "drop cast"
4\ lines 89-98: the authors are really recommended to give brief details of how their NPs were prepared. For example, laser parameters (wevelength, frequency, pulse energy and duration of ablation, etc.) should be given. Any reader who wants to see such details should see them in the Experimental section (not in ref. [20-29])
5\ Lines 64-71. Somewhere in the Introduction, the authors are recommended to mention other works on PLAL -generated NPs for hydgrogen generation. The following review should be referred to and some important reports in the review can be picked for comparison with the results reported by the authors:
33 (10.1016/j.cogsc.2021.100566
The above review specifically mentions NPs produced by lasers for hydrogen generation. That's why the authors should really pay attention to it.
6\ Line 130: must be "cast" (rather than "casted")
7\ Lines 138-139: The authors refer to their previous work [29] when they mention their Cu(OH)2 NPs . However, in work [29], they reported on CuO NPs. Please explain the difference and why in the present manuscript, the authors conclude that the NPs are Cu(OH)2 rather than CuO.
8\ Figure 2, panels (e) and (f): why do these panels have their labels in different locations? If this is ONE figure, then all labels should be placed in a similar way.
Author Response
Reviewer #4
The manuscript is of interest for the journal but it needs revision prior to being considered for acceptance. After revision, it can be reconsidered. Below are comments and suggestions to help the authors improve it:
Author reply: we would like to thank this Reviewer for his/her very competent and unbiased review process.
1\ English should be improved here and there, for example, in Conclusions: (a) synthesis ... produce ; (b) an high ; (c) produces the formation. Also, in Abstract, line 20: probably should read "consists in reducing..."
Author reply: thank you for these suggestions, we carefully checked the manuscript for English errors.
2\ Line 448: must be Electrochem. Commun. (not commun.)
Author reply: the error has been corrected, thank you.
3\ Line 97: instead of "drop casted", it should be "drop cast"
Author reply: the change has been made, thank you.
4\ lines 89-98: the authors are really recommended to give brief details of how their NPs were prepared. For example, laser parameters (wevelength, frequency, pulse energy and duration of ablation, etc.) should be given. Any reader who wants to see such details should see them in the Experimental section (not in ref. [20-29])
Author reply: to cope this comment we add more information in the lines 98-101.
5\ Lines 64-71. Somewhere in the Introduction, the authors are recommended to mention other works on PLAL -generated NPs for hydrogen generation. The following review should be referred to and some important reports in the review can be picked for comparison with the results reported by the authors: Current Opinion in Green and Sustainable Chemistry 33 (2022) 100566. doi:10.1016/j.cogsc.2021.100566
The above review specifically mentions NPs produced by lasers for hydrogen generation. That's why the authors should really pay attention to it.
Author reply: Thank you for the tips, I add this interesting reference as new Ref 26.
6\ Line 130: must be "cast" (rather than "casted")
Author reply: the change has been done, thank you.
7\ Lines 138-139: The authors refer to their previous work [29] when they mention their Cu(OH)2 NPs . However, in work [29], they reported on CuO NPs. Please explain the difference and why in the present manuscript, the authors conclude that the NPs are Cu(OH)2 rather than CuO.
Author reply: in the reference [29] the “as prepared” NPs by PLAL were drop cast onto interdigitated electrode and then annealed in a carbolite horizontal oven at 400 °C for 45 min in synthetic air (80% N2 and 20% O2 v/v %, relative humidity (RH) < 3%). In that condition the as prepared NPs, that are formed by a mixture of Cu2O/CuOH and Cu(OH)2 (on the surface) are converted into CuO. In the present work we did not performed the annealing then the NPs are formed by Cu(OH)2 highly hydrophilic.
8\ Figure 2, panels (e) and (f): why do these panels have their labels in different locations? If this is ONE figure, then all labels should be placed in a similar way.
Author reply: we changed the Figure 2 according to the reviewer suggestions. Thank you for the tips.

Round 2
Reviewer 3 Report
Several images are already published in a previous publication of the same authors (https://doi.org/10.3390/mi13060963) lile 6 SEM images in figure 1, XPS spectra presented in Figure 2,the results of GP, Pt, Pd, and PtPd presented in Figures 3 and 4.
Authors should omit the duplicated results.
Author Response
Several images are already published in a previous publication of the same authors (https://doi.org/10.3390/mi13060963) lile 6 SEM images in figure 1, XPS spectra presented in Figure 2, the results of GP, Pt, Pd, and PtPd presented in Figures 3 and 4.
Authors should omit the duplicated results.
Author reply: to address this reviewer's comment we changed Figure 1a,b,c,e,f,g with new FE-SEM images. To do this we prepared and analyzed the new samples, obtaining almost the same results. We also performed new XPS and electrochemical characterizations of the systems, reproducing the previous results perfectly. We would like to point out that the Figures of the two papers (https://doi.org/10.3390/mi13060963 and the actual ones) are completely different. ONLY some XPS and electrochemical characterizations are similar, NOT the entire Figures. As mentioned in the first answer, we conceived this work as a continuation of the previous work published in Micromachines, adding and comparing the results on Cu(OH)2. For this reason, we have resumed, obviously quoting them in the text, some measures. We deemed it necessary to report some previously published data in order to be able to make a quantitative comparison in the same graphs and on the same scales of the various quantities. The aim was to facilitate the reading and comparison of the data obtained with Pt, Pd PtPd with those relating to Cu(OH)2. Removing these measurements, and we emphasize measurements, not whole Figures, would mean distorting the meaning of our work.
Only from this comparison emerges the originality of the result reported in this work and an advance with respect to the previous literature. We believe that the comparison is solid and effective for the reader of this article only through the direct comparison expressed in the Figures shown and eliminating the images indicated by the referee would make the message partial and ineffective.
Moreover, we rewrote the Equations 2-5 using the specie H2+ads instead of Hads. This new intermediate is based on recent literature results reported in the new references [26 and 41]. For these new aspects, the paper has to be considered original in various parts and not a duplication.
On the other hand, we do not consider anything shocking, as happens in the literature, for example in 90% of the works on batteries, photovoltaics, semiconductors, etc., that different authors report their data obtained in different phases of the research in the same final graph with their own data and /or of literature to show the improvement of their data.
We invite the referee to carefully evaluate our point of view and we thank him/her for the comment which, however, led us to improve the work.

Reviewer 4 Report
The authors still need to improve their Reference List:
1\ Ref.57, must be: Li, Z.; Jalil, S.A.; .....
2\ Ref.49, must be: Dietze, E.M.; .....
3\ Ref.48, must be: Nakamura, J.; Cambell, J.M.; ....
4\ Ref. 26, third author must be: Kulinich, S.A.
5\ Ref.63: all authors should be presented as Surname, Initials
6\ There might be more mistakes in the ref. list, which is why the authors MUST check it carefully. If they wish to be respected (when cited) by others, the authors should cite others with respect, avoiding mixing names/surnames and so on.
Author Response
The authors still need to improve their Reference List:
1\ Ref.57, must be: Li, Z.; Jalil, S.A.; .....
2\ Ref.49, must be: Dietze, E.M.; .....
3\ Ref.48, must be: Nakamura, J.; Cambell, J.M.; ....
4\ Ref. 26, third author must be: Kulinich, S.A.
5\ Ref.63: all authors should be presented as Surname, Initials
6\ There might be more mistakes in the ref. list, which is why the authors MUST check it carefully. If they wish to be respected (when cited) by others, the authors should cite others with respect, avoiding mixing names/surnames and so on.
Author reply: we would like to thank the reviewer for his/her check of the references. Now the references have been corrected. Moreover, we corrected the citations of the Ref. [10] in the main text of the manuscript at lines 65 and 282.

Round 3
Reviewer 3 Report
Accept